# 40 Years after the Registration of Acyclovir: Do We Need New Anti-Herpetic Drugs?

**DOI:** 10.3390/ijms23073431

**Published:** 2022-03-22

**Authors:** Anna Majewska, Beata Mlynarczyk-Bonikowska

**Affiliations:** 1Department of Medical Microbiology, Medical University of Warsaw, Chałubińskiego 5, 02-004 Warsaw, Poland; anna.majewska@wum.edu.pl; 2Department of Dermatology, Immunodermatology and Venereology, Medical University of Warsaw, Koszykowa 82a, 02-008 Warsaw, Poland

**Keywords:** HSV, VZV, CMV, antivirals, resistance mechanisms

## Abstract

Herpes simplex virus types 1 and 2 HSV1 and 2, namely varicella-zoster VZV and cytomegalovirus CMV, are among the most common pathogens worldwide. They remain in the host body for life. The course of infection with these viruses is often asymptomatic or mild and self-limiting, but in immunocompromised patients, such as solid organ or bone marrow transplant recipients, the course can be very severe or even life-threatening. Unfortunately, in the latter group, the highest percentage of infections with strains resistant to routinely used drugs is observed. On the other hand, frequent recurrences of genital herpes can be a problem even in people with normal immunity. Genital herpes also increases the risk of acquiring sexually transmitted diseases, including HIV infection and, if present in pregnant women, poses a risk to the fetus and newborn. Even more frequently than herpes simplex, congenital infections can be caused by cytomegalovirus. We present the most important anti-herpesviral agents, the mechanisms of resistance to these drugs, and the associated mutations in the viral genome. Special emphasis was placed on newly introduced drugs such as maribavir and brincidofovir. We also briefly discuss the most promising substances in preclinical testing as well as immunotherapy options and vaccines currently in use and under investigation.

## 1. Introduction

Chemotherapeutic agents used to treat viral diseases are classified in three categories: virucides, antivirals, and immune response modifiers. Virucides are used to prevent the transmission of viruses [1]. Examples include organic solvents, detergents, nanoparticles, and ultraviolet light [1,2,3]. Antivirals (drugs and prodrugs) have critic role in the therapy of viral diseases and spread of viruses. Some of them are used to prevent infection or to prevent disease outbreak (so-called suppressive or pre-emptive therapy). Most of the medications have a narrow spectrum of action and can inhibit only one or a few closely related viruses [1]. Immune response modifiers augment the host response to infection by intensifying the cell-mediated immunity and humoral response [1,4]. So far, some drugs with immunomodulatory activity have been approved in therapy of viral diseases, such as interferons, inosine pranobex, and zinc oxide [4,5,6,7,8]. A separate group are corticosteroids, which, on the one hand, due to their immunosuppressive effect, may worsen the course of viral diseases and, on the other hand, are sometimes used to reduce inflammation. Such adjunctive treatment may support the controlling of serious infections, e.g., herpesviral encephalitis and herpetic infections of the skin, as well as conjunctivitis (depends on the physician’s decision) [5,8]. Antivirals are compounds that inhibit the formation of new viruses by interference with certain steps of viral replication. Understanding the replication cycle and viral targets is essential for the development of new antiviral drugs [1,7]. Despite the fact that so many compounds and substances have been developed, tested, and applied, the treatment of diseases caused by herpes viruses and the prevention of infections require urgent improvement.

### Herpesviruses, Taxonomy, and Pathogenicity

*Herpesvirales* consists of numerous species. Most of them are potentially infectious for a wide range of animal species (including insects, fish, molluscs, reptiles, birds, and mammals). The family *Herpesviridae,* based on their genome sequence and biological characteristics, is divided into three subfamilies. Nine human herpesviruses are known so far [9].


Order: HerpesviralesFamily: HerpesviridaeSubfamily: Alphaherpesvirinae


➢Genus: SimplexvirusSpecies: Human alphaherpesvirus 1 (known as herpes simplex virus type 1; HSV-1)Human alphaherpesvirus 2 (known as herpes simplex virus type 2; HSV-2)➢Genus: VaricellovirusSpecies: Human alphaherpesvirus 3 (known as varicella-zoster virus; VZV)


Subfamily: Betaherpesvirinae


➢Genus: CytomegalovirusSpecies: Human betaherpesvirus 5 (known as cytomegalovirus; CMV)Genus: Roseolovirus➢Species: Human betaherpesvirus 6AHuman betaherpesvirus 6BHuman betaherpesvirus 7


Subfamily: Gammaherpesvirinae


➢Genus: LymphocryptovirusSpecies: Human gammaherpesvirus 4 (known as Epstein–Barr virus; EBV)➢Genus: RhadinovirusSpecies: Human gammaherpesvirus 8 (known as Kaposi’s sarcoma-associated herpesvirus; KSHV) [10].

The herpesvirus genome consists of a non-segmented, linear, double-stranded DNA (dsDNA) molecule. It is made up of, depending on the virus species, 125 to 240 kbp and encodes from 35 to over 200 proteins. The virion consists of an icosahedral nucleocapsid surrounded by a lipid envelope. The replication cycle of *Herpesviruses* takes place in the host nucleus. The best known and described herpesvirus is HSV-1. It encodes more than 90 proteins, seven of which are essential for DNA replication: origin-binding protein UL9, single-stranded (ss) DNA-binding protein ICP8 (UL29), the heterotrimeric primosome encoded by the UL5 (helicase), UL52 (primase), UL8 (non-catalytic subunit) genes (UL8/UL5/UL52), DNA polymerase, UL30, and polymerase processivity factor UL42 [11]. 

The first step of DNA replication involves the circularization of the double-stranded DNA (dsDNA). The circular molecules function as templates for DNA synthesis. The UL9 protein acts as a DNA replication initiator and binds to DNA in the ori region. UL9 has ATPase and helicase activity that unwind the origins of DNA replication. In the next step, the helicase–primase complex (HP) is recruited to viral replication forks. The HP complex (composed of UL5, UL8, and UL52 proteins) has 5′ to 3′ helicase, ATP-ase, primase, and DNA-binding activities. UL5 unwinds the double-stranded dsDNA into two single strands (replication fork) before DNA synthesis. Helicase uses energy derived from the hydrolysis of adenosine triphosphate (ATP) to unwind DNA. Catalytic properties of the complex are retained in a subcomplex consisting of UL5 and UL52. The UL8 mediate protein is required for ICP8 to stimulate helicase/primase activity. It seems that UL8 is important for the nuclear import of UL5 and UL52. The next step is the synthesis of short RNA sequences complementary to the parental strand of DNA. RNA fragments act as primers allowing for the initiation of DNA synthesis. The viral protein UL30 with the function of DNA polymerase and the UL42 are responsible for the DNA synthesis. Virus thymidine kinase (UL23), ribonucleotide reductase (UL39/UL40), and deoxyuridine triphosphatase (UL50) are involved in the synthesis of deoxyribonucleotides triphosphate (dNTPs) [1,12,13,14]. 

Antiviral drugs can interfere with a specific stage of the replication cycle. With herpesviruses, inhibitors of the polymerase and terminase complex, and the benzimidazole inhibitor of UL97 CMV kinase are registered for use so far. The methylenecyclopropane analogue that inhibits polymerase, the ATP competitive inhibitor, and helicase-primase inhibitors are now under advanced research. Antiviral drugs approved for use in herpes infections in humans are shown in Figure 1. 

Currently registered anti-herpes-viral drugs can control infections caused only by HSV, VZV, and CMV [13,14,15]. Studies on effective and specific immunoprophylaxis have been conducted for years but only vaccines against VZV (chickenpox and herpes zoster) are currently available [14,15,16].

Human herpesviruses are ubiquitous. Infections are widespread all over the world, with clinical manifestation ranging from asymptomatic, mild, and self-limiting to severe and life-threatening. *Herpesviruses* can cause persistent cutaneous lesions, serious organ infections (esophagitis, meningitis, severe neurological sequelae, pneumonia, and liver inflammation), and disseminated disease in immunocompromised hosts (solid organ recipients, hematopoietic stem cell (HSC) transplant recipients, immunodeficiency virus (HIV)-infected individuals). Some species (e.g., HSV, VZV, and CMV) are responsible for congenital infection and/or neonatal infection [15,16,17]. *Herpesviruses* can produce lytic or latent infections. During latency, the viral genome is kept in the host cell (e.g., in the sensory ganglia, B or T lymphocytes, macrophages, and lymphocytes depending on the species) and can reactivate to cause recurrent outbreaks. Anti-herpes compounds do not eradicate viruses in a latent state [16,17,18]. *Human gammaherpesviruses* (EBV and KSHV) have an oncogenic nature and are causative agents of approximately 2% of all human cancers [17,19]. Although herpesviruses are host-specific, some of them might potentially cross the species barriers. It was shown that *Cercopithecine herpesvirus* 1 (known as B virus disease; CeHV-1) infecting macaque monkeys, *Equid alphaherpesvirus* 1, or poultry-infected *Gallid alphaherpesvirus* 2 (known as Marek’s disease virus; MDV) and *Suid alphaherpesvirus* 1 (known as pseudorabies virus; PRV causes Aujeszky’s disease) are potentially epizootic [9,10].

## 2. Antiviral Drugs

The antiviral era started with iododeoxyuridine (IDU), the first effective antiviral agent. IDU is a pyrimidine analogue synthesized in 1959 as an anticancer drug. In 1963, IDU became the first antiviral agent used topically to treat herpes simplex keratitis [7]. Trifluridine (TFD) was synthesized in the early 1960s as a cytotoxic antitumor drug. Due to lower selectivity and greater toxicity than other nucleoside analogues, its use is remarkably limited to eye infections caused by HSV-1 [7,20]. At about the same time, vidarabine (VDR, Ara-A) was synthesized. The drug was used intravenously but because of its toxicity, its use was significantly limited to the local treatment of eye HSV infections [1,7]. Gertrude Elion, who received a Nobel Prize in Physiology or Medicine Laureate (1988), contributed to the development of acyclovir (ACV), the first of the second-generation nucleoside analogues, a highly selective inhibitor of alpha-herpesviruses [21]. Acyclovir was discovered in early 1970s. Preclinical investigation brought the drug to clinical studies in 1977. ACV was first approved for use as an antiviral agent in 1982 and has been still in antiviral armamentarium [21,22,23].

Since then, many potentially antiherpesviral compounds have been synthesized [14,24,25]. Some of them have been approved and marketed [1,7,22,26,27,28]. Up to date, there are no specific, highly effective, and safe antiviral drugs against *Human betaherpesviruses* and *Human gammaherpesviruses.*

### 2.1. Nucleoside Analogues 

Nucleoside analogues are highly selective inhibitors which target the virus-encoded DNA polymerase (DNA pol). Its antiviral effect results from the inhibition of the viral DNA synthesis in the mechanism of the competitive incorporation of the deoxyguanosine triphosphate (dGTP) into the DNA chain. Nucleoside analogues reduce symptoms, viral shedding and the frequency of outbreaks. They can be used as a suppressive therapy, preemptive therapy, and risk-adapted prophylaxis. Nucleoside analogues include acyclovir (ACV), ganciclovir (GCV), penciclovir (PCV), and derivatives of these drugs (prodrugs) with better bioavailability, such as valacyclovir (VACV), valganciclovir (VGCV), and famciclovir (FCV) [15,18,22,29].

The mechanism of action of nucleotide analogues compared to other anti-herpes-viral drugs is shown in Figure 2 [30,31,32,33] and Figure 3 [30,31,34,35,36,37].

Acyclovir is a synthetic, acyclic analogue of guanosine in a side chain of which, instead of the traditional cyclic sugar residue, a 2-hydroxyethoxymethyl acyclic side chain is introduced. It is available in topical, oral, and intravenous formulations. ACV is the first-line treatment of herpes simplex (HSV-1, HSV-2, and VZV). Acyclovir and valacyclovir may be used as a suppressive therapy to prevent oral and genital recurrences of a disease caused by HSV-1 and HSV-2 [7,15,18,28]. Lately, suppressive oral antiviral therapy after treatment of acute infection in infants with neonatal herpes has been proposed [38]. The course of HSV infection in newborns is very severe, with mortality rates of up to 60% in untreated infants [39]. ACV therapy substantially improves the survival of neonates and decreases the incidence of both disseminated disease and central nervous system (CNS) infection. Survivors usually suffer from cutaneous recurrences. Infants with disseminated or CNS infection are at high risk of neurodevelopmental complications. Considering that the estimated incidence rate for neonatal herpes is comparable to other perinatally acquired diseases (syphilis or HIV infection), for which screening assays are available, the standard of care for management of neonatal HSV disease should be improved [38,40,41].

ACV is safe and well-tolerated. The bioavailability of ACV is only 15–30%. The synthesis of valacyclovir, the L-valine ester of ACV (bioavailability of about 54%) overcame the problem of poor oral ACV bioavailability [15,22].

In 2009, the United States Food and Drug Administration (US FDA) approved a topical cream containing a combination of 5% ACV and 1% hydrocortisone (AHC) for the prevention and treatment of cold sores. The AHC cream reduces the recurrence of ulcerative and nonulcerative herpetic lesions, and shortens healing time with early treatment compared with acyclovir 5% cream and placebo cream [42,43].

Valacyclovir, penciclovir, and famciclovir are less popular in treatment and prevention, although they are still found in the armamentarium of anti-alpha-herpes-viral drugs. All of the above mentioned nucleoside analogues require phosphorylation to triphosphate forms before binding to and inhibiting the viral DNA polymerase. The first phosphorylation occurs *via* a thymidine kinase (TK) enzyme encoded by the *UL23* (HSV) and *ORF36* (VZV) genes, which is not essential for the viral replication but works as a drug activator. Next, the monophosphate form of a drug is converted by host cellular kinases to its active triphosphate form and as a substrate of the viral DNA polymerase (encoded by the HSV *UL30* and VZV *ORF28* genes) is incorporated into the DNA at its 3′ terminus, preventing further chain elongation. Acyclovir and its derivatives act only in virus-infected cells. The benefit of this pathway is low toxicity and potent selectivity [16,28,29].

Penciclovir is an acyclic guanine derivative. The spectrum of its activity and mechanism of action is similar to ACV. When given orally, it is poorly absorbed, which was compensated for by the development of a prodrug of PVC, namely famciclovir (oral bioavailability of 77%). Famciclovir is effective against HSV-1 and HSV-2 (as a standard therapy and in suppression of recurrent outbreaks), and against VZV [16].

Brivudin (BVDU) is a thymidine nucleoside analogue originally synthesized in 1976 and available on the market since the 1980s in several European countries (not in the U.S.). Among the herpesviruses, VZV proved highly sensitive to BVDU. It was demonstrated that BVDU was more effective than acyclovir in the topical treatment of HSV-1 infections (i.e., herpes labialis and herpetic keratitis), although inhibition of HSV- 2 is much less efficient. The main indication for BVDU use is the oral (125 mg daily for 7 days) treatment of VZV infections (i.e., herpes zoster in immunocompromised patients). The half-maximal effective concentration (EC_50_) of brivudin is 0.0024 μg/mL; thus, compared to acyclovir (EC_50_ 4.64 μg/mL), BVDU is more than 100-folds more superior in potency of BVDU against VZV replication than acyclovir in cell culture experiments. It was also demonstrated that BVDU effectively reduced local symptoms (healing of lesions) and prevented acute pain in postherpetic neuralgia, which is a common complication of herpes zoster [26,44,45,46]. In spite of this, BVDU is not widely used. Over the years, ACV and VACV are market leaders in the treatment of alpha-herpes-viral infections. The limited activity of BVDU against HSV-2 is a significant limitation of the drug. Although, perhaps due to the market monopolization of ACV and its analogues, the antiviral properties of BVDU have not been thoroughly investigated [47]. BVDU is generally well-tolerated and adverse effects are similar to those of acyclovir (most often nausea and headache). The main metabolite of brivudin is bromovinyluracil, thus BVDU simultaneously administrated with a cytostatic drug used in the treatment of cancer, namely fluorouracil (5-FU), may enhance its toxicity. Brivudine resistance occurs in thymidine kinase-deficient strains, thus cross-resistance to ACV can be common. Single-nucleotide (nt) exchanges, resulting in amino acid (aa) substitutions, were observed within the thymidine kinase (ORF 36) and/or DNA polymerase (ORF 28). There are no reports on clinical isolates with BRVD resistance that developed during therapy but BVDU-resistant HSV-1 clones can be readily selected in vitro during replication in the presence of a drug. Amino acid substitutions with Ala168Thr within the nucleoside-binding site causes BRVD resistance but aciclovir/penciclovir/foscarnet/cidofovir retains susceptibility [48,49]

Ganciclovir is a synthetic, acyclic nucleoside analogue of deoxyguanosine. GCV is structurally similar to ACV but introducing an additional hydroxyl group (-OH) has expanded the drug activity. Besides alpha-herpesviruses, GCV inhibits the replication of CMV. It was showed that GCV inhibits EBV; HHV 6, 7, and 8; and hepatitis B virus (HBV) replication, as well. In the clinical practice, it is the drug of choice to treat the CMV infection [50,51,52,53,54,55]. GCV is indicated to treat cytomegalovirus retinitis as a suppressive therapy, used for the prevention of CMV disease in transplant recipients. GCV is first phosphorylated by the viral protein kinase (encoded by the CMV *UL97* gene) and then diphosphorylated by cellular kinases. GCV is an inhibitor of viral DNA polymerase (encoded by the CMV *UL54* gene) [18,50,51]. When orally administered, it is absorbed in only 5–10%. Since 2004, GCV is used only parenterally. Due to its the low bioavailability, it was replaced with valganciclovir. GCV has a less favorable safety profile compared to ACV. Adverse effects include hematotoxicity and nephrotoxicity [18].

Valganciclovir is a L-valyl ester salt of GCV. The bioavailability of the drug is 60%, which is about 10 times higher than GCV. VGCV is well-absorbed and rapidly metabolizes to GCV. The indications for use are CMV retinitis, CMV disease prevention in solid organ (heart, kidney, or kidney-pancreas) transplant (SOT) patients, and CMV colitis or esophagitis in HIV-infected patients (off-label). VGCV is available in the form of tablets and powder for oral solution [56].

Infections caused by CMV are a major problem in high-risk group of patients, such as the solid organ recipients and allogeneic hematopoietic stem cell recipients. CMV disease may be associated with life-threatening complications. Viremia is a strong marker to predict post-transplantation risk of CMV disease and thereby acts as a good biomarker for making decisions about pre-emptive prophylaxis (PET) [28,37]. VGCV and GCV use as a PET in solid organ recipients may reduce the rate of late CMV infection [57,58]. In patients after allogenic HSC transplantation, prophylaxis of CMV infection, and reactivation, strategies include, besides universal prophylaxis and preemptive therapy, also risk-adapted prophylaxis. GCV is effective but too toxic for HSC recipients. VGCV causes myelosuppression, thus its use is significantly limited [59].

In a case of resistance to ganciclovir CMV infection, foscarnet (FOS) is used off-label, but it has many limitations that will be mentioned later. Anyway, it is a necessity for more drugs that are both safe and effective against CMV. The most needed are new possibilities for the control CMV viremia and prevention of CMV disease in SOT and HSC transplant patients. Recently, the therapeutic options have been expanded to include the new drugs described below, such as letermovir, maribavir, and brincidofovir [51,52,60].

VGCV may be beneficial for EBV-related diseases [61]. It has been shown that VGCV effectively reduced mucosal replication of HHV-8 in a randomized clinical trial as detected by a PCR assay [62]. It has been also demonstrated that human herpesvirus 6 and 7 antigenemia often occur in patients early after lung transplantation. GCV or VGCV treatment of CMV infection can be effective against the concomitant HHV 6 and 7 antigenemia, although CMV prophylaxis does not prevent the infections with these viruses [54].

Resistance to nucleoside analogues is a significant clinical issue. Viruses have a high grade of multiplication, hence a high rate of mutation. The selection of resistant variants can be related to the prolonged use of antiviral drugs or treatments with suboptimal doses. However, drug-resistant strains have also been isolated in the absence of a known history of antiviral treatment [15,18]. In immunocompetent individuals, the prevalence of nucleoside analogues’ resistance in HSV is still low (below 1%), but in immunocompromised HSV-infected patients, it varies from 2.5% to over 30%. The resistance rate depends on the degree of immunosuppression and underlying illness (the highest in HSC recipients under prolonged antiviral therapy) [27,44,63,64,65]. 

Infection caused by the HSV-resistant strain should be considered in patients not responding clinically to appropriate doses of antiviral therapy, previously treated with ACV, or patients with recurrent disease [50]. Resistance to nucleoside analogues may be due to the substitutions, insertions, or nucleotides deletions, and is attributed to frameshift mutations within the gene (HSV *UL23* and VZV *ORF36*) encoding viral thymidine kinase. The most significant regions involved in the activity of an enzyme are: (1) the ATP binding site, (2) nucleoside binding site, and (3) region responsible for the three-dimensional structure of the active site the cysteine at codon 336 [66,67]. HSV resistance to ACV is mostly related to mutations in the *UL23-*the thymidine kinase gene (90–95% of cases) and less frequently to mutations in the *UL30-*viral DNA polymerase gene (5–10% cases) [68,69]. Mutations occurring within the *UL30* gene may induce resistance not only to nucleoside analogues (ACV) but also to foscarnet and nucleotide analogues (CDV) [63,66,70]. The TK is non-essential for viral replication, thus TK-deficient and TK-altered mutants can replicate successfully. It was noticed that the TK gene plays a substantial role in α-herpesvirus virulence. In studies on animal models, it has been shown that TK-producing strains are more virulent [71]. 

The emergence of VZV isolates resistant to ACV has not been studied in immunocompetent patients with varicella or herpes zoster. Published information show that therapy of VZV infections with ACV or VACV in immunocompetent hosts is not associated with the selection of resistant strains. The problem of resistance in immunosuppressive patients also remains not fully understood. It has been reported that 27% of hemato-oncological patients (including HSC recipients) with persistent VZV infections had mutations that may be associated with ACV resistance. However, it should be pointed out that there are a lack of contemporary studies dealing with this problem. Most VZV isolates resistant to ACV have been isolated from children infected with HIV [5,45,70,71]. 

Long-term exposure to GCV and the use of suboptimal doses (suboptimal plasma concentrations) may contribute to the selection of CMV-resistant strains [18]. It was shown that the problem of resistance to ganciclovir may affect 1.5–10% of organ transplant recipients and 0–14.5% of HSC recipients. Resistance to GCV is associated mainly (over 95% of cases) with a mutations in the CMV *UL97* gene, which encodes a kinase essential for the phosphorylation of the drug. Mutations in the CMV *UL54* gene encoding DNA polymerase are less frequent but may result in cross-resistance to GCV, FOS, and CDV [18,52,72]. The most common mutations conferring resistance to nucleoside analogues are shown in Table 1 [66,67,68,69,72,73,74,75,76,77,78,79,80,81,82,83,84,85,86,87].

### 2.2. Nucleotide Analogues

Cidofovir (CDV) is an analogue of cytosine monophosphate. As a monophosphate nucleotide analog, CDV is phosphorylated to its active, diphosphate form. It does not require phosphorylation by thymidine kinase, thus it can inhibit the replication of viruses that do not produce this enzyme. CDV is a competitive inhibitor of viral DNA polymerases, inhibits the incorporation of deoxycytidine triphosphate (dCTP) into viral DNA by viral DNA polymerase, and finally inhibits the DNA elongation (Figure 2 and Figure 3). The principal indication for CDV is to treat the following CMV infections: retinitis among adult patients with AIDS, CMV infections caused by strains resistant to GCV and FOS, and severe infection (progressive mucocutaneous infections in immunocompromised patients). Other indications for the use of CDV are infections caused by HSV and VZV resistant to ACV and/or FOS. CDV also acts as an inhibitor of other viruses: adenoviruses (HAdVs), polyomaviruses, human papillomaviruses (HPVs), and orthomyxoviruses [18,28,64,70]. CDV has limited bioavailability (5 to 22%) and is therefore given intravenously. Nephrotoxicity is the most common adverse effect. In the kidney cells, the concentration of cidofovir is about 100 times higher than in other tissues, thus CDV is routinely administered with probenecid. Other adverse effects are neutropenia and myelosuppression. CDV application is currently limited due to the toxicity of the drug and patients should be monitored during and after the treatment [18,52,70,72].

The limited bioavailability of CDV following oral administration contributed to the synthesis of brincidofovir (BCDV), which contains a synthetic, acyclic monophosphate nucleotide analogue (cidofovir) conjugated to a lipid (3-hexadecyloxy1-propanol) *via* a phosphonate group. Conjugation with a lipid molecule improves BCDV delivery to the target cells and also significantly reduces the nephrotoxicity compared to CDV [52,88].

BCDV is orally available and has a long intracellular half-life. The advantage of the drug is a broad-spectrum of in vitro confirmed activity against herpesviruses (VZV, HSV, CMV, EBV, and HHV6), polyomaviruses, adenoviruses, and papillomaviruses [88,89]. In June 2021, BCDV was approved by the FDA only to treat smallpox and was indicated for the treatment of adults and pediatric patients weighing at least 13 kg [90]. It was also a candidate for the CMV infection therapy. Despite promising preclinical data, the results of most clinical trials evaluating its efficacy were, however, disappointing [89,91].

Due to its unfavorable toxic profile, the use of CDV is limited. The problem of clinical drug resistance may be not common, although it is also not extensively studied. Mutations associated with resistance to CDV are mapped in DNA polymerase. So far, a few CDV-resistant CMV isolates have been described. Some of them are also resistant to GCV as a result of mutations within the *UL54* gene but may remain susceptible to FOS. In CMV *UL54*, some amino acid changes may be related to only FOS resistance. Multidrug-resistant CMV variants (to GCV, FOS, and CDV) were also described. CMV mutants resistant to CDV have been isolated from patients under GCV therapy but more often were selected in vitro as a result of long-term exposure to CDV. It was shown that amino acid substitutions in CMV strains exposed to BCDV are associated with BCV, GCV, and CDV cross-resistance [18,66,69,92,93,94,95].

### 2.3. Analogue to Pyrophosphate

Foscarnet (FOS) is a non-nucleoside analogue to pyrophosphate. FOS is a structural mimic of the anion pyrophosphate that selectively inhibits the pyrophosphate binding site on viral DNA polymerase and finally prevents the incorporation of nucleotides into the growing DNA strand (Figure 2 and Figure 3). It is one of the few drugs that have a broad spectrum of antiviral activity (HSV, VZV, EBV, HHV6, HBV, and HIV [22,70,96]. Foscarnet is indicated to treat HIV-infected patients with CMV retinitis that do not tolerate GCV or as a salvage therapy for those who have drug-resistant CMV infection and fail GCV. FOS is used as an off-label in certain other CMV diseases, such as esophagitis, colitis and outer retinal necrosis, and in VZV infection [18,22]. As a second-line drug, it is also used for the treatment of infections caused by the ACV-resistant HSV and CMV strains resistant to GCV. Important limitations of the FOS therapy are significant adverse effects such as nephrotoxicity, e.g., interstitial nephritis, acute renal tubular necrosis, and electrolyte derangement, although it seems that FOS causes minimal myelosuppression. Due to its poor bioavailability, FOS is not orally available and given through intravenous administration [18,22,96].

Resistance to foscarnet is associated with mutations at the pyrophosphate binding site of virus DNA polymerase [44]. Amino acid substitutions conferring resistance of HSV-1 and CMV to FOS are located in the genes *UL30* and *UL54*. It has been shown that those most resistant to FOS HSV isolates contain single-base substitutions in the conserved regions of II, III, VI, or VII, and in non-conserved regions (from I to VII). Some mutations localized in regions II and VII of the DNA polymerase can contribute to resistance to both ACV and FOS. In a case of CMV, amino acid substitutions conferring resistance to FOS are mainly distributed in the palm, fingers, and NH2-terminal domains of the *UL54* DNA polymerase, whereas mutations associated with cross-resistance to FOS and GCV are located in the fingers domain [18,92,97,98].

### 2.4. Quinazoline Derivative

Letermovir (LMV) is a novel compound that represents a new class of non-nucleoside CMV inhibitors and 3,4 dihydro-quinazoline-4-yl-acetic acid derivatives. Letermovir’s target is the terminase complex composed of proteins and the gene products *UL51, UL56,* and *UL89* (Figure 3). This compound inhibits CMV replication in the stage of DNA maturation and packaging into a capsid. As an inhibitor of CMV, LMV is about 1000 times more active than GCV. When co-administrated with GCV or CDV, it shows an additive or synergistic effect in vitro. LMV is available in an oral (bioavailability is 35%) and intravenous form, is well-tolerated, and does not cause nephrotoxicity or a myelosuppressive effect. In November 2017, LMV was approved by the U.S. FDA for prophylaxis of CMV infection and cytomegaloviral disease in adult CMV-seropositive allogeneic hematopoietic cell recipients [18,99,100,101]. It is a very important indication because CMV-seropositive patients undergoing allo-HCT are at increased risk of CMV infection. CMV reactivation occurs in 80% patients with no antiviral CMV prophylaxis [57,83,101]. It is the only indication for LMV. However, its role as a CMV infection prevention in solid organ recipients has been studied [57,100]. Four clinical trials are now being conducted and two of them (both in phase 3) have evaluated the letermovir as a potential therapeutic option to prevent cytomegalovirus infection and disease in kidney transplant recipients (NCT04129398 and NCT03443869) [102,103]. The efficacy of LMV for the prevention of CMV infection and disease in adult lung transplant recipients (NCT05041426) [104] and heart transplant recipients who are at risk for cytomegalovirus disease (NCT04904614) [105] has been investigated as well [106,107]. In in vitro and in clinical trials, some mutations connected to LMV resistance have been demonstrated [18,108,109,110,111,112]. Changes occur mainly within the *UL56* gene and less commonly within the *UL89* and *UL51* genes. The first case of clinical LMV resistance was reported in 2016 [109]. Based on in vitro experiments and observation during clinical studies, it was assessed that LMV may possess low genetic barriers to resistance. Mutations associated with resistance to LMV arise relatively easily and quickly [108,109,110,111]. Resistance to LMV has been found as early as in 30 days into LMV exposition [109,111]. This feature may be a significant limitation of the long-term use of LMV as prophylaxis of cytomegaloviral disease but synergism with GCV is a beneficial feature of the drug [106].

### 2.5. Benzimidazole Inhibitor

Maribavir (MBV) is a benzimidazole L-riboside ATP competitive inhibitor of the CMV *UL97* kinase (Figure 3). It is the first drug that works by preventing the activity of the UL97 protein. It binds specifically to the serine/threonine kinase, which mediates one of the final stages of viral replication and inhibits and encapsulation and escape newly formed virions from the nucleus of an infected cell. Maribavir is available only in the oral form [18,113].

This compound showed high antiviral potency in vitro and favorable properties during preclinical and early clinical testing [114]. On 23 November 2021, maribavir was FDA-approved to treat post-transplant cytomegalovirus infection that does not respond to available antiviral treatment. MBV has a beneficial toxicity profile [115]. The most common side effects include taste disturbance, nausea, diarrhea, vomiting, and fatigue. However, maribavir may reduce the antiviral activity of GCV and VGCV [114]. UL97 is a kinase that is not essential for the virus replication. CMV strains with deletions of the entire *UL97* gene or in which UL97 kinase activity has been abrogated remain viable but replicate less efficiently. It was demonstrated that mutations in the *UL97* gene (V353A, L397R, T409M, and H411L/N/Y) appear in vitro as a result of MBV selection pressure and confer moderate-to-high levels (nine-fold reduction in susceptibility to over 200-folds) of MBV resistance. MBV-resistant variants (T409M, H411Y, or C480F) were found in strains isolated from recurrent CMV infections during maribavir therapy. The shortest MBV exposure associated with an appearance of the listed resistance mutation was 6 weeks. Mutations in another CMV gene, namely *UL27*, are related to low-grade (two to three-fold increase) resistance to MBV and have not been identified so far during clinical use of the drug. Mutation C480F can cause the highest level of MBV resistance (224-fold increased) and low-grade (2.3-fold reduction in susceptibility) GCV cross-resistance but the growth of C480F variants is impaired. The most common mutations conferring resistance to antivirals other than nucleoside analogues are shown in Table 2 [18,52,69,70,83,87,89,92,93,94,95,99,108,113,116,117,118,119,120,121].

## 3. Compounds with Potential Use in the Treatment of Herpesvirus Infections

### 3.1. Helicase-Primase Inhibitors (HPIs)

HPIs constitute a new class of antiviral drugs with a new mechanism of action. It was noticed that the helicase-primase (HP) complex is an attractive target [7,11,27]. HPIs are novel small molecules that can be applied to control alpha-herpes-viral infections [26,44,122,123].

The first analogue targeting HP complex (T157602) was reported in 1998. The next compounds were synthetized at the beginning of the 21th century. It seems that HPIs are highly effective against skin and vaginal herpes infection in vitro and in vivo, have low cytotoxicity in vitro, and are well-tolerated in mice and humans [12,13,123,124]. Unlike ACV and its derivatives, HPIs do not require phosphorylation to the active form by viral kinase and show activity in uninfected cells [11,122,125]. The mechanism of action of helicase-primase is shown in Figure 2 [126,127,128,129].

Hitherto, three classes of herpesvirus HPIs were developed:➢2-amino-thiazolylphenyl derivatives (BILS 179 BS),➢thiazole urea (BAY 57-1293; pritelivir), and➢oxadiazolylphenyl type (ASP2151, amenamevir) [12].

BILS 179 BS was identified by screening inhibitors of DNA unwinding [11,50]. It inhibits DNA-stimulated ATP hydrolysis and inhibits HSV helicase-primase activity by preventing DNA release during translocation. In the plaque formation assay (PFA), it has been shown that BILS 179 BS decreases HSV DNA replication of wild type HSV strains or ACV-resistant mutants cultured in cell culture [12]. The EC_50_ of BILS 179 BS for HSV-1 and HSV-2 is 0.08–0.10 µM and 0.010–0.011 µM, respectively. For ACV-resistant HSV-1 and HSV-2 strains, the EC_50_ values are 0.13 µM and 0.09 µM [123]. BILS 179 BS was also tested on a mice model infected with HSV-1 and HSV-2. When taken orally, it was more effective than ACV. An interesting finding is that the infection was successfully restricted when initiation of treatment was delayed up to 65 h after infection [124]. Thus, HSV-infected mice treated with BILS 179 BS or its derivative (BILS 45 BS) have fewer and smaller cutaneous and genital lesions than a control group of mice receiving a placebo [125].

Another promising helicase-primase inhibitor in preclinical testing is IM-250{(S)-2-(2’,5’-difluoro-[1,1’-biphenyl]-4-yl)-N-methyl-N-(4-methyl-5-(S-methylsulfon-imidoyl)thiazol-2-yl)acetamide}. Animal studies have shown that the substance penetrates exceptionally well into target tissues including the nervous system. In contrast to currently used drugs, an effect on latent infection and virus reactivation is observed, which persists after withdrawal. Such results give hope that HSV infection will become curable in the future [130].

Since then, two other HPIs have emerged as promising drug candidates, namely pritelivir (BAY 57–1293) and amenamevir (ASP2151), and both are under advanced research.

Pritelivir (PTV, formerly named AIC316 or BAY 57-1293), acts by inhibiting the ssDNA-dependent ATPase activity of the helicase-primase complex. The compound exhibits antiviral activity against HSV-1 and HSV-2 isolates in vitro and in animal models (mice and guinea pigs). The PTV EC_50_ for HSV-1, HSV-2, and VZV is 0.016–0.042 µM, 0.032–0.12 µM, and 0.038–0.10 µM, respectively [123,124]. The compound is given orally or intraperitoneally, is highly effective against HSV-1 latent infection in mice, and seems to be superior to ACV, VACV, FCV, or GCV. PTV applied topically, with good results, reduced recurrent skin lesions in mice infected with HSV-2 [13,123,124]. It has been demonstrated that when administered orally to mice after dermal inoculation with HSV-2, it was more effective than ACV [13,124]. Similarly, when given topically, it was effective in preventing ocular disease and encephalitis in mice after corneal scarification and infection with HSV-1 [124]. Wald and co-workers conducted extensive studies on the effects of PTV in HSV-2-infected individuals. In a study of adults (healthy men and women ≥ 18 years of age) who were seropositive for HSV-2 and had a history of genital herpes, it was shown that PTV reduced the rates of genital HSV shedding. In patients treated with a placebo, the median log10 number of HSV DNA copies was 5.1. After oral administration of PTV, the median log10 decreased in a dose-dependent manner to 4.5 with 5 mg of PTV daily, 3.6 with 25 mg daily, 2.4 with 75 mg daily, and 3.6 with 400 mg weekly. The effect of PTV on the clinical manifestations of genital infection with HSV-2 was also assessed as a percentage of the days on which participants had genital lesions. The study lasted for a period of 28 days. Symptoms of genital infection were noted on 9.0% of the days with the placebo and 12.5% with 5 mg of pritelivir, 3.5% with 25 mg, 1.2% with 75 mg, and 1.2% with 400 mg of PTV [131]. Among the adults who experienced a frequently recurring genital HSV-2 infection, the use of PTV compared with VACV resulted in a lower percentage of swabs with HSV detection (2.4% vs. 5.3%). Genital lesions were presented on 1.9% and 3.9% of the days in the PTV and VACV cohort, respectively [132]. Effective pritelivir therapy in recurrent genital herpes in an allogenic peripheral blood stem cell transplant recipient infected with acyclovir-resistant HSV-2 was also described [133]. PTV directly inhibits the replication of HSV, limits the intracellular viral load, and prevents the spreading of the virus into a new cell. PTV therapy results in limited HSV-2-spread from neurons into epithelial cells of the genital tract and limits cell-to-cell-spread within a genital ulcer [134]. Further studies are needed to assess the utility of PTV in a specific group of patients. Recruitment to a new study concerning the efficacy and safety of PTV tablets for treatment of ACV-resistant mucocutaneous HSV infections in immunocompromised individuals (NCT03073967) is currently underway [135]. It has been demonstrated that HPIs’ resistance mutations located in the helicase or primase genes can be selected in vitro. All resistance-mediating mutations to PTV identified so far in vitro are located at a single amino acid position in the viral *UL52* primase (amino acid 906) and viral *UL5* helicase (the conserved helicase motif; amino acids 341–355). To date, no resistant variants with an amino acid change in *UL8* have been identified [136,137,138].

Amenamevir (AMV; ASP2151) is an oxadiazolylphenyl-containing helicase-primase inhibitor with activity against human alpha-herpesviruses [12,137,139]. Besides HSV (HSV-1 EC_50_ 0.047 and HSV-2 EC_50_ 0.028 μM), AMV inhibits the replication of VZV (EC_50_ 0.047 µM) [12]. Amenamevir showed better efficacy in comparison with VACV in the treatment of HSV skin lesions in immunocompromised mice [140]. In a phase 2 study, it was demonstrated that once-daily AMV and VACV administered twice a day for 3 days appear to be effective (expressed in time to lesion healing) and safe options for treatment of recurrent genital herpes episodes [141]. This compound remains active against mutants with the kinase thymidine defective [45,142]. A combination with ACV and other nucleoside analogues demonstrated a synergistic/additive effect against HSV and VZV in vitro. Such combined therapy may by an advantageous option for threating severe infections, e.g., encephalitis and infections in immunosuppressed individuals [12,45,142]. The oral bioavailability of AMV is 86%. It is claimed that AMV is safe for the patient and well-tolerated but the toxicity profile is not thoroughly investigated. Renal function needs to be monitored as N-acetyl-β-glucosaminidase and α1-microglobulin is excreted in the urine. Based on the clinical trials results, it seems that AMV does not cause serious side effects in both renally normal and impaired patients. It was demonstrated that AMV cures herpes zoster and prevents postherpetic neuralgia, as in 2017 it was approved in Japan (oral dosage of 400 mg for 7 days) for therapy of herpes zoster [45,143]. In Japan, AMV has been used to treat at least 1,240,000 individuals with a herpes zoster diagnosis. In these groups of patients, the frequency of adverse reactions could be lower than in people treated with the other anti-herpes-virals.

It was demonstrated that resistance to HPIs may be associated with mutations located in *UL5* or *UL52* [12]. As mentioned before, mutations in *UL52* causing HSV-1 resistance to HPIs are located within the N- and C-domains. Mutations in the N-domain (S364G and R367H) combined with mutations in *UL5* (G352V and M355I) confer a 3000-fold increase of the resistance to AMV [45,144].

Pacreau et al. demonstrated a very low natural polymorphism in the HP complex of VZV which displays the lowest nucleotide variation within its genome (estimation as 0.05–0.06%) of all the human herpesviruses [145]. Today, one mutation conferring VZV resistance to AMV has been identified. Sequence analysis revealed the N336K change in the *ORF55* helicase of VZV, corresponding to the N342K change in the *UL5* helicase of HSV-1 in the virus exposed to the presence of increasing concentrations of AMV in cell culture. There is a need to monitor the resistance of VZV strains to AMV [45,145].

### 3.2. Methylenecyclopropane Analogue

Filociclovir (FCV, MBX-400; formerly cyclopropavir) is a small molecule, the second-generation of the nucleoside analogue and promising compounds for the treatment of CMV infections. This 2′-deoxyguanosine analogue containing a methylenecyclopropane moiety has successfully completed phase I human clinical safety studies [146,147,148]. It blocks CMV replication about 10-folds more effectively than GCV. The EC_50_ of FCV is ~0.50 µM. The high efficiency was also demonstrated in experiments on animal models [148,149]. FCV is now being developed for the treatment of CMV-related disease in immunocompromised patients. It has a favorable oral bioavailability. After administration to healthy volunteers, no serious adverse events were observed [149,150]. The most common mutations conferring resistance to helicase-primase inhibitors and filociclovir [119,150] are shown in the Table 3 [12,119,137,138,142,144,145,150].

### 3.3. Other Selected Substances in Preclinical Studies

Recently published results of cell culture studies on amidinourea derivatives of low molecular-weight moroxidine analogues are promising. They showed a lack of cytotoxicity and activity against HSV. It was also shown that the tested substances have a different mechanism of action than the nucleotide analogues. They act on the early stages of virus replication [151]. MBZM-N-IBT1-[(2-methyl benzimidazole-1-yl) methyl]-2-oxo-indolin-3-ylidene] amino] thiourea is also in preclinical testing. MBZM-N-IBT1 was shown to inhibit HSV-1 replication by affecting viral protein synthesis. Specifically, it reduces the synthesis of HSV-1 proteins, namely UL9, gC, and ULC8, and interferes with the production of proteins, namely ICP8, ICP27, UL42, UL25, UL15, and gB [152]. Another possibility is to act on cellular enzymes involved in HSV replication. The inhibitor of dihydroorotate dehydrogenase (DHODH), an enzyme involved in pyrimidine synthesis MEDS433 in cell culture studies, showed activity against HSV-1 and 2, and acted synergistically with acyclovir [153]. Studies on the effect of the ellagitannins on HSV have also shown promising results. Ellagitannins are a type of polyphenols and are esters of a monosaccharide, usually glucose and gallic or hexahydroxydiphenyl acids. They can be obtained from certain plants, especially pomegranates and berries. They are potent antioxidants, exhibit anti-inflammatory effects, and are also being studied for their anticancer and antimicrobial properties. Ellagitannins such as castalagin, vescalagin, and grandinin have been found to inhibit HSV replication and act synergistically with ACV. The mechanism of action is likely a complex and is not fully understood. Hydrolysable tannins such as chebulagic acid and punicalagin have been found to inhibit HSV entry into cells. In the case of the in vivo action of these substances, not only direct antiviral effects may be of importance but also anti-inflammatory, antioxidant, and immune response-modifying effects [154,155,156].

## 4. Vaccines

Despite similarities in the structure and replication cycle, there are important differences in the way VZV, HSV, and CMV spread in the host and the extent to which they can evade the immune response. This has implications not only for the clinical course of individual infections but also for the development of effective prophylactic and therapeutic vaccines. These differences, to some extent, explain why a vaccine against VZV has been available for quite a long time and why, despite years of research, there are still no vaccines available for CMV and HSV.

### 4.1. VZV

After chickenpox, significant immunity usually develops and reactivation of VZV is relatively difficult. The occurrence of reactivation and symptoms of zoster is associated with additional exposure of the virus to the host immune system and further improvement of immunity, thus relapses of zoster are relatively rare and mainly affect immunocompromised individuals. The severe course of chickenpox and zoster in patients with cellular immune disorders indicates a major role for this type of immune response in controlling VZV infections. On the other hand, the effective action of varicella-zoster immune globulin (VZIG) in exposed individuals suggests that the humoral response is also important. Passive prophylaxis of VZV infection (VZIG) has long been available. It is still recommended, i.e., for preterm newborns or unvaccinated immunocompromised individuals exposed to VZV infection [157]. The production of neutralizing antibodies against viral glycoproteins E and B, the gH/gL complex, and antibody-dependent cell-mediated cytotoxicity (ADCC) has been observed after both VZV infection and the attenuated vaccine [158].

The high mortality rate from chickenpox in children with leukemia spurred intensive vaccine research in the 1970s of the 20th century. The result of this research was the development in 1974 in Japan of a vaccine based on a live strain of virus attenuated by passage on guinea pig embryo cultures (vOka strain). The vaccine proved to be effective and safe also in healthy individuals. In 1995, it was approved by the FDA for one-dose prophylactic vaccination. However, some infections were reported in vaccinated children. It was not until the recommendations were changed and children were vaccinated with two doses that the incidence of chickenpox decreased by over 95%. The vaccine is available both as a stand-alone preparation (Varivax and Varilvax) and in combination with measles, mumps, and rubella vaccines (ProQuad and Priorix-Tetra) [159].

Subsequent clinical trials in people over 60 years of age have shown that administration of the vOka strain at a dose 14 times higher than that of chickenpox prophylaxis reduces the risk of zoster by 51%. The product (Zostavax) was approved by the FDA in 2006 for use in people over 50 years of age for the prevention of herpes zoster [159].

A newer vaccine containing recombinant viral glycoprotein E and adjuvant AS01B— consisting of two immunostimulants, namely saponin and the monophosphoryl lipid A (3-O-desacyl-40-MPL)-toll like receptor TLR4 antagonist—has proven even more effective in preventing herpes zoster. Clinical trials have shown a 97% efficacy for two doses of this preparation administered to individuals 50–70 years of age. The vaccine (Shingrix) was approved by the FDA in 2017 [160].

### 4.2. CMV

CMV infection affects most of the general population but is usually well-controlled by the immune system. The problem arises in virtually only two situations. The first is significant immunosuppression, e.g., in people with AIDS or in organ and especially bone marrow transplant recipients. Transplant recipients are often already infected with CMV but reactivation of the virus or reinfection are also dangerous for them. The second situation is infection in pregnant women and the associated risk of vertical transmission. CMV is considered the most common pathogen transmitted vertically, occurring, depending on the study group, in 0.2% to 2% of live births. The highest risk is primary infection during pregnancy but reinfection (affecting about 1–2% of seropositive women) or reactivation of the virus can also be dangerous. Congenital CMV infection can have serious and permanent consequences, such as mental retardation or hearing loss [161,162].

CMV has evolved numerous mechanisms to evade the host immune response. Among others, US7 and US8 bind to TLR (toll-like receptor) 3 and TLR4, and UL94 interacts with the metabolic pathway cGAS (Cyclic GMP-AMP synthase)-MITA (mediator of IRF3 activation) involved in antiviral response. This leads to inhibition of the production of type 1 interferons and other pro-inflammatory cytokines. Another mechanism is the downregulation of HLA-IA expression on the surface of infected cells, which impairs their recognition by T cells [163,164].

The production of antibodies against the surface proteins involved in viral entry, such as glycoprotein gB (UL55), gH (UL75), gL (UL115), gM(UL100) gN (UL73), and gO (UL74), has been shown to play an important role in the successful immune response to CMV infection. Natural resistance to infection is specifically associated with the production of neutralizing antibodies against the pentamer complex gH/gL/UL128/UL130/UL131. The stimulation of specific cellular immunity by pp65, pp150, and pp50/52tegument proteins, and immediate early proteins may also be important [165].

The first vaccines based on live attenuated CMV strains, i.e., the AD169 and Towne vaccines in clinical trials, proved to be ineffective in preventing the infection, although in the case of Towne, a beneficial effect on the course of disease caused by CMV in transplant patients was observed. It is speculated that the reason for the lack of a prophylactic effect of the vaccines may have been the lack of the pentamer complex gH/gL/pUL128-131 on the surface of the attenuated virus. Currently, in phase II of clinical trials, there is a vaccine V160 based on the AD169 strain genetically modified to restore the pentamer complex and with both IE1 and UL51 genes introduced. The vaccine is well-tolerated and has been shown to stimulate the production of neutralizing antibodies in similar titers to natural infection as well as cellular responses. The immune response against the pentamer complex is also expected to be stimulated by another vaccine in clinical trials. The vaccine mRNA-1647 contains a self-replicating m-RNA encoding a pentamer complex and gB. Several DNA vaccines are also under investigation, most of which encode the pp65 teugment protein, among others [166,167]. CMV vaccines recently in clinical studies are shown in Table 4 [168,169,170,171,172,173,174,175,176,177,178,179,180,181,182]

### 4.3. HSV

As with CMV, the most severe course of HSV infection is seen in immunosuppressed individuals and neonates. However, in contrast to CMV infection, frequent symptomatic recurrences of herpes can also be a problem in adults in the general population. In addition, genital herpes is a factor that increases the risk of HIV transmission. Studies using mathematical models have shown that the use of a prophylactic vaccine in the United States with an efficacy of as low as 50% would reduce, by 2050, annual new HSV infections by 58%, incidence by 60%, and seroprevalence by 21%. With a therapeutic vaccine, these rates would be lower in 2050 by 12%, 13%, and 4% [183]. Another study on the potential use of a vaccine to prevent the acquisition and recurrence of HSV infection with 75% efficacy in sub-Saharan Africa found that it would reduce HIV incidence by 30–40% after 20 years [184].

Unfortunately, the remarkable ability of HSV to evade the immune response and its greater ease of reactivation than VZV or CMV infection makes the creation of an effective vaccine a difficult task that has yet to be accomplished. The fact is that the natural immune response is not sufficient to fully control the infection, as evidenced by the frequent occurrence of herpes relapses even in immunocompetent individuals. An important role in the induction of an immune response is played by glycoprotein D, located on the surface of the virus and involved in its entry into cells. Many tested vaccines were effective in inducing the production of neutralizing antibodies mainly against D-glycoprotein; however, HSV spreads directly through intercellular junctions, which means that blocking antibodies often does not work. To some extent, this may explain the insufficient efficacy of the prophylactic vaccines tested in the 1990s of the 20th century containing recombinant gD and gB (Chiron) or gD (Simplirix™, GSK). Interestingly, Simpilrix showed 58% efficacy in preventing genital HSV-1 but not HSV-2 infection in seronegative women [185]. Furthermore, the vaccines induced mainly neutralizing antibody production, whereas ADCC-associated antibodies probably play a more important role in an immune response against HSV [186]. However, the virus is able to evade this form of immune system response, as well. Glycoprotein E binds to the Fc fragment of IgG antibodies, blocking their action, including ADCC. Other numerous mechanisms by which HSV evades the immune response include the production of proteins that block interferon production and action (e.g., ICP0, ICP34.5, and UL46), and block the complement system by glycoprotein C, which binds to the complement component C3b [187,188]. Subunit, RNA and DNA vaccines are in preclinical testing to induce the formation of neutralizing antibodies against gD2 gE and gC, which may allow for blocking some of the above mechanisms and more effective action. An mRNA vaccine containing these antigens is about to enter phase I clinical trials [189,190].

A separate problem is that HSV vaccines are often more effective in animal studies than in later clinical trials. A frequently used model is the mouse and guinea pig, which, unlike the mouse, can have recurrent herpes and asymptomatic genital mucosal shedding. However, these models do not always give adequate results. In addition to the differences in immune response, this may also be due to the frequent use of different criteria for assessing vaccine efficacy in animals and humans (such as evaluating the occurrence of genital lesions in humans versus their severity and duration in animals). Another animal model used is the macaque, which has an immune system more similar to humans but is expensive and does not develop genital lesions after HSV infection [191]. Cotton rats were also tried, showing a higher correlation with clinical trials of the Simplirix vaccine compared to guinea pig studies [192].

Due to the large number of people already infected, therapeutic vaccine research is also important. In 2013, the results of phase II clinical trials of a vaccine based on 32 different HSV peptides (fragments of gD and other surface proteins, among others) in combination with both recombinant human heat shock protein HSP 70 and adjuvant QS-21 Stimulon (Agenus’ HerpV) were announced. In the study group, viral shedding was reduced by 15% and the viral load by 34% compared to the placebo group. A phase II clinical trial of a therapeutic vaccine (GEN-003, Genocea) containing two recombinant viral proteins, namely infectious cell protein-4 (ICP4, essential for the HSV replication protein involved in the activation of early E and late viral genes), gD, and adjuvant Matrix-M (saponin combined with cholesterol and phospholipids), was completed in 2017. A dose-dependent reduction in episodes of asymptomatic HSV-shedding and symptomatic recurrences was demonstrated. The number of symptomatic days was reduced by up to 41% compared to the placebo group [193].

DNA vaccines have also been studied. This group included monovalent (gD2) and bivalent (gD2 and UL46 tegument proteins) vaccines in combination with the adjuvant Vaxfectin (Vical). Phase 1/2 clinical trials were completed in 2015. The bivalent vaccine was significantly more effective than the monovalent vaccine and there was a decrease in the viral shedding, viral load, and number of days with clinical symptoms (the latter by 51%) compared to pre-vaccination values. Unfortunately, there were no statistically significant differences from the placebo group, in which all of the above values also decreased [194].

Another study of a COR-1 vaccine (Admedus)—which contains two plasmids, namely one encoding the entire D2 glycoprotein and the other encoding a truncated D2 protein in combination with a ubiquitin sequence—is expected to lead to rapid degradation by the proteasome to peptides, to subsequent antigen presentation in the context of HLA class I, and to subsequent stimulation of a CD8+ T cell-mediated immune response. In the phase I/IIa clinical trials, there was a decrease in the viral shedding and a reduction in the symptoms relative to the baseline but not relative to the placebo. In both of the studies described above, the placebo groups were small and it is likely that studies in larger groups could clarify the questions that arose [195,196].

Vaccines containing the entire virus may generate a more diverse immune response, which could theoretically increase the chance of effective action. The first such vaccine contained a virus devoid of the H-glycoprotein, which was supposed to prevent the spread of the virus between cells and limit its activity to a single life cycle. Despite very promising results in an animal model (guinea pigs), phase II clinical trials showed no effect on the number or course of herpes recurrences in patients [196].

Another vaccine, HSV 529 (Sanofi Pasteur), containing a virus devoid of the DNA replication proteins UL5 and UL29, is currently in phase I/II clinical trials. Preliminary results have shown the safety of the vaccine. Neutralizing antibody titers increased by at least four-folds in 78% of HSV-1 and HSV-2 seronegative study participants but there was no significant increase in HSV-1 and 2 seropositive patients. Further studies are planned to administer HSV 529 along with gD2 and both the capsid antigens UL19 and UL25 adjuvanted with glucopyranosyl lipid A [197].

The ΔNLS vaccine (Rational Vaccines) is officially entering the first phase of clinical trials. It contains a live virus attenuated by the deletion of the gene encoding ICP0. The vaccine was previously tested without FDA approval in a small group of HSV-infected patients. The investigators reported a reduction in the recurrence rates in this group [198,199].

HSV vaccines currently under investigation are shown in Table 5 [158,190,194,195,197,198,199,200,201,202,203,204,205,206].

## 5. Conclusions

In summary, the currently available anti-herpes drug groups are the following.

Viral DNA polymerase inhibitors such as:
-nucleoside analogues (ACV, GCV, PCV, VACV, VGCV, and BVDU FCV);-nucleotide analogues (CDV); and-analogue to pyrophosphate (FOS).Helicase-primase inhibitors (AMV in Japan).CMV UL97 kinase inhibitor (MBV).CMV terminase complex inhibitor (LMV).

Clinical trials include helicase-primase inhibitors (PTV), a new nucleoside analogue FCV and other indications for drugs such as AMV, MBV, LMV, or BCDV.

Newly introduced anti-herpes drugs often have better bioavailability and greater potency. They are also often active against herpesvirus strains resistant to older generation drugs.

However, it is worth noting that despite the development of new drugs, acyclovir is still the most commonly used drug in cases of HSV and VZV infections susceptible to this drug. There has long been a search for a drug that would have at least the same safety profile as ACV and would work against both CMV infection and resistant cases of HSV and VZV. In the coming years, it remains to be seen whether drugs such as LMV, MBV, or FCV are the solution to the problem in the case of CMV and helicase-primase inhibitors in the case of HSV and VZV.

The problem that remains is viral resistance, which also develops with new drugs. Some of them, such as MBV, have a low threshold for developing drug resistance. Infections with drug-resistant virus strains mainly affect immunocompromised patients with the most severe course of disease.

Additionally, it is worth mentioning that there are still no drugs available or even in clinical trials that would allow for the complete elimination of herpesviruses from the body of an infected person.

In view of the high prevalence of HSV, CMV, and VZV infections, prevention is of great importance. Currently, only vaccines and passive imunoprophylaxis against VZV are available. Advanced clinical trials are underway for effective vaccines against HSV and CMV infections. It is hoped that these vaccines will become available in the future.

## Figures and Tables

**Figure 1 ijms-23-03431-f001:**
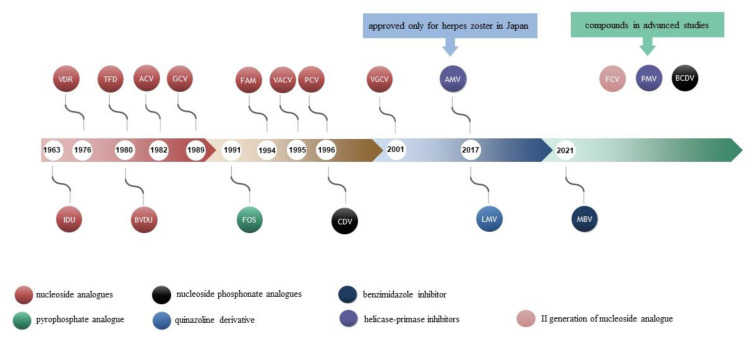
Antiviral drugs approved for use in herpesviruses infections in humans and potentially antiviral compounds.

**Figure 2 ijms-23-03431-f002:**
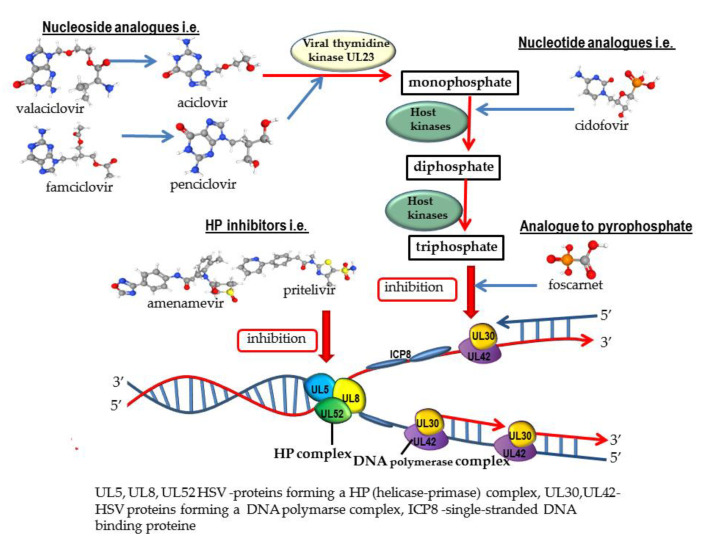
The mechanism of action of nucleoside analogues, nucleotide analogues, and analogues of pyrophosphate and HP-inhibitors on HSV-1.

**Figure 3 ijms-23-03431-f003:**
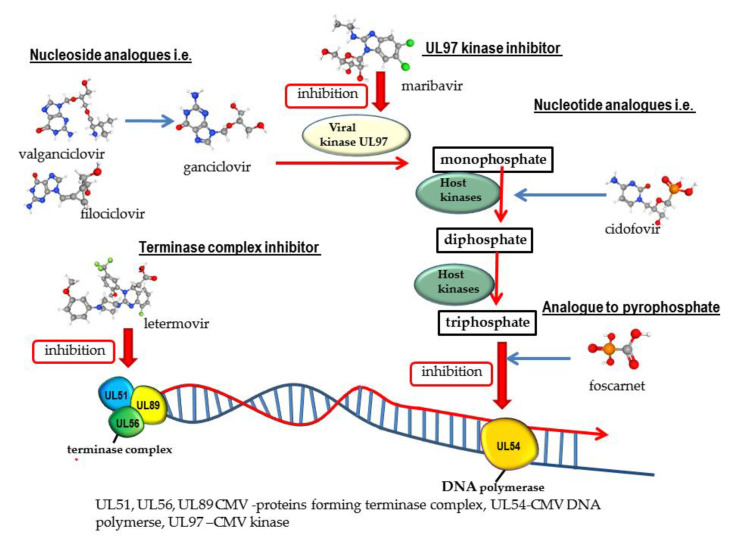
The mechanism of action of nucleoside analogues, nucleotide analogues, and analogues of pyrophosphate, the terminase complex inhibitor, and the UL97 kinase inhibitor on CMV.

**Table 1 ijms-23-03431-t001:** Nucleoside analogues. Structure and the most common ACV and GCV/VGCV resistance-conferring mutations.

Drug Name/IUPAC Name	Chemical Formula
ACICLOVIR (ACV) 2-amino-9-(2-hydroxyethoxymethyl)-1H-purin-6-one	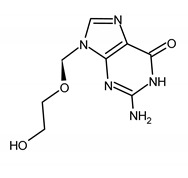
**Mutations conferring resistance to ACV**	**Ref.**
HSV-1 *UL23*: G200C/S/D, R261C, R281STOP, L298P, A189V, A93, V204G, Y53C/D/H, Y53stop, D55N, G56S, P57H, K62N/R, A174P, H58R/Y, Q259stop, Q67stop, Y172S, C336Y, D162N/T, L242P, L170P, W88stop, C171stop, R176stop, S181N, R216S, R220C, R221H, G246V, T287M, R41H, Q261R, P84S, M121L, E83K, P84L, Y87H, G ins to 7Gs nt.430-436, stop codon at AA 224, A ins at nt. 438, stop codon at AA 227, E257K, S263stop (del nt. 781–793), L288stop, E39G, L208F, del C548-55399, R51W, and G59P *UL30*: R700G/M, A719V/T, A605V, S724N/S729N, V813M, G841C, G850I, L773M, Y941H, V573M, L1007M, I1028T, E188K, E222K, M501I, M553L, G901V, del G432-438, S599L, F733C, S775N, T821M, R881C, I619K, V715S, N820NS, R1047L, V621S, and H1228D	[66,67,68,69,72,73,74,75,76,77,78,79,80,81]
HSV-2 *UL23*: M183stop, R220C, R221C/H, D137stop, Q222C, G61W, G201D, E84K/G, S169P, T288M, R51W, R177W, R217H, E39G, Q105P, R271V, add G nt. 433–439, del G nt. 439–440, del C nt. 467, and G59P *UL30*: Y577H, E678G, D785N, D307N, Q619R, A724T/V, Q732R, N820S, T843A, T844I, G846A, and A915V	[66,68,69,72,77,78,81]
VZV *ORF36:* S179N, W225R, W225stop, E48G/L, G24E/R, K25R, A37stop, I38stop, R54stop, E59, T86, C90stop, L92, H97R, D129N, R130Q, C138R, R143G/K, L154P, A163STOP, V171STOP, V194STOP, Y206stop, C231STOP, T256A/M, L298stop, Q303stop, L332P, N334stop, D338stop, ins. nt. 412-3, and del nt. 641 *ORF28*: E512K, K662E, R665G, V666L, D668Y, V680A, A684T/N, Q692R, A773V, N779S, I804T, G805C, R806S, M808V, L809S, V855M, and M874I	[68,69,79,82,83]
VALACICLOVIR (VACV) 2-[(2-amino-6-oxo-1H-purin-9-yl)methoxy]ethyl (2S)-2-amino-3-methylbutanoate	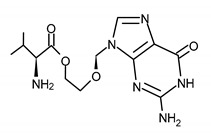
PENCICLOVIR (PCV) 2-amino-9-[4-hydroxy-3-(hydroxymethyl)butyl]-1H-purin-6-one	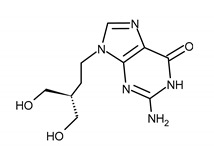
FAMCYCLOVIR (FCV) [2-(acetyloxymethyl)-4-(2-aminopurin-9-yl)butyl] acetate	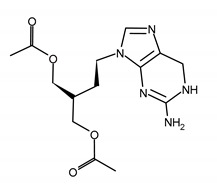
GANCICLOVIR (GCV) 2-amino-9-(1,3-dihydroxypropan-2-yloxymethyl)-1H-purin-6-one	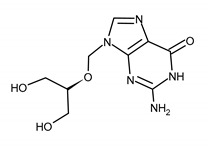
VALGANCICLOVIR (VGCV) [2-[(2-amino-6-oxo-1H-purin-9-yl)methoxy]-3-hydroxypropyl] (2S)-2-amino-3-methylbutanoate	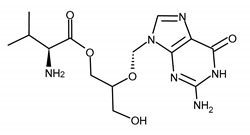
**Mutations conferring resistance to GCV**	**Ref.**
CMV *UL54*: D301N, N408D, N410K, F412C, D413E, L501F/I, T503I, K513R/E, P522A, Q578H, V781I, L802M, A809V, G841A, A834, V787L, and del. 524 *UL97*: del595-596, C592G, H520Q, K599T, M460V/I, V466G, A594V/E/G/T/P, C603W/R, L405P, and L595F	[84,85,86,87]
BRIVUDIN (BVDU) [(E)-5-(2-bromovinyl)-2′-deoxyuridine]	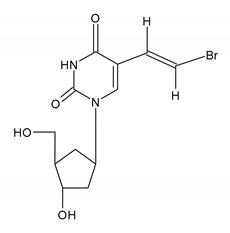

**Table 2 ijms-23-03431-t002:** Nucleotide analog (CDV), analogue to pyrophosphate (FOS), quinazoline derivative (LMV), and benzimidazole inhibitor (MBV). Structure and the most common resistance-conferring mutations.

Drug Name/IUPAC Name	Chemical Formula
CIDOFOVIR (CDV) [(2S)-1-(4-amino-2-oxopyrimidin-1-yl)-3-hydroxypropan-2-yl]oxymethylphosphonic acid	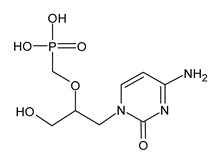
**Mutations conferring resistance to CDV**	**Ref.**
HSV *UL30:* R700M, G841C, G850I, L773M, Y941H, V573M, L1007M, I1028T, K960R, and N408K i V812L CMV *UL54:* K805Q, D542E, D301N, N408D, N410K, F412C/V, D413A/E, L501I, T503I, L516R, I521T, P522A, L545S, D588N, E756K, V812L, T813S, G841A, A987G, K513E/R, S676G/S Y751H, and del 981–982 VCV *ORF28:*K662E, V666L, D668Y, L767S, N779, G805C, and R808V	[18,69,70,83,92,93,94]
BRINCIDOFOVIR (BCDV) [(2S)-1-(4-amino-2-oxopyrimidin-1-yl)-3-hydroxypropan-2-yl]oxymethyl-(3-hexadecoxypropoxy)phosphinic acid	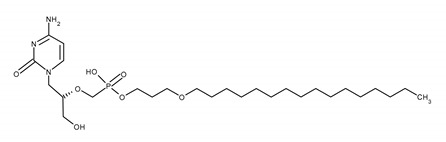
**Mutations conferring resistance to BCDV**	**Ref.**
CMV *U54:* N408K, V812L, D413Y, E303D, E303G, D542E, and E303G+ V812L	[52,89,95]
FOSCARNET (FOS) phosphonoformic acid	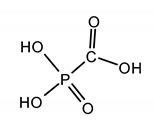
**Mutations conferring resistance to FOS**	**Ref.**
HSV-1 and HSV-2 *UL30: *V715G, S724N/S729N, Y941H, I922N, I619K, V715S, and A719T CMV *UL54:* Q697P, V715M/S, A719T, T700A, M715V, V781I, V787L, L802M, A809V, V812L, T821I, Q578H, N495K, D588E, and E756D/N/Q T838A. V715M, S290R, T813S, A834, G841, del. 981-982, VCV *ORF28:*E519K, K662E, R665G, V666L, D668Y, L767S, G805C, R806S, M808V, L809S, V855M, Q692R, and N774	[18,68,69,92,93,94,95]
Letermovir (LMV) 2-[(4S)-8-fluoro-2-[4-(3-methoxyphenyl)piperazin-1-yl]-3-[2-methoxy-5-(trifluoromethyl)phenyl]-4H-quinazolin-4-yl]acetic acid	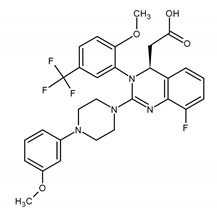
**Mutations conferring resistance to LMV**	**Ref.**
CMV *UL56*: C25F, L328V, V231L, (C25F +V231L) V236M, (V236M+L257I+M329T), V236A(V236L+L257I), L241P, L257F, C325Y, C325W, C325F, A365S, R369G, R369M, R369S, T4270, T4189, T4257, T4237, and T4217 *UL89*: N320H, N329S, D344E and T350M, especially when coexisting with Q204R, E237D, F261L, and M329T in *UL56* (*UL89* D334E +*UL56* E237D) *UL51* P91S (*UL51* P91S + *UL5* R369M)	[52,87,94,99,108]
Maribavir (MBV) (2S,3S,4R,5S)-2-[5,6-dichloro-2-(propan-2-ylamino)benzimidazol-1-yl]-5-(hydroxymethyl)oxolane-3,4-diol	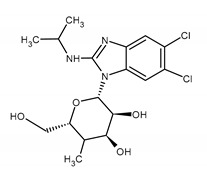
**Mutations conferring resistance to MBV**	**Ref.**
CMV *UL97*: T409M, H411Y, F342Y, F342F, C480F, V466G, P521L, V356G, D456N C480R, Y617del., M460V/I, H520Q, A594V, L595S*, *C603W H411N, H411L, L347R, V353A *UL27:* L335P, 480F	[99,116,117,118,119,120,121]

**Table 3 ijms-23-03431-t003:** Compounds with potential use in the treatment of herpesvirus infections: helicase-primase inhibitors and second-generation of nucleoside analogue. Structure and the most common resistance-conferring mutations.

Drug Name/IUPAC Name	Chemical Formula
PRITELIVIR (PTV) N-methyl-N-(4-methyl-5-sulfamoyl-1,3-thiazol-2-yl)-2-(4-pyridin-2-ylphenyl)acetamide	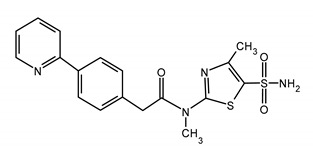
**Mutations conferring resistance to PTV**	**Ref.**
**HSV-1***UL5:*K356T; *UL52*: A899T, UL52: A899T, and UL5: K356T	[131,138]
AMENAMEVIR (AMV) N-(2,6-dimethylphenyl)-N-[2-[4-(1,2,4-oxadiazol-3-yl)anilino]-2-oxoethyl]-1,1-dioxothiane-4-carboxamide	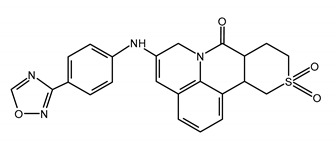
**Mutations conferring resistance to AMV**	**Ref.**
**HSV-1***UL5:*G352V+M355I; *UL52*: R367H + S364G; and HSV-2 *UL5:* K355N + K451R	[12,132,136]
**VZV*** OFR55:* R66H and *OFR6:* N939D	[139]
FILOCICLOVIR (FCV) 2-amino-9-[(Z)-[2,2-bis(hydroxymethyl)cyclopropylidene]methyl]-1H-purin-6-one	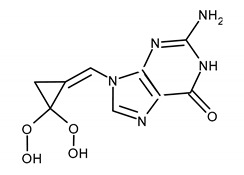
**Mutations conferring resistance to FCV**	**Ref.**
**CMV***UL97:* M460V, M460I, H520Q, C592G, A594V, and C603W (cross-resistance with ganciclovir in all cases)	[119,143]

**Table 4 ijms-23-03431-t004:** HCMV vaccines.

Name of Vaccine (Developer)	Antigen/Other Vaccine Components	Type of Vaccine	Research Stage	Results	References
**V160 (Merck)**	whole virus and AD169 with restored pentamer complex	Live attenuated	Phase 2 completed	Phase 1: well-tolerated and immune response similar as in natural infection. Phase 2: 42.4% preventive efficacy in seronegative women of reproductive age	[168,169]
**gB/MF59 (Sanofi Pasteur)**	gB with MF59 adjuvant	Recombinant subunit	Phase 2 completed	Safe and immunogenic, 43% preventive efficacy in seronegative adolescent girls, and was found to be insufficient. Neutralizing antibody production in a small group of seronegative transplant recipients from seropositive donors	[170,171]
**CMVPepVax** **(City of Hope)**	pp65 fused to a natural tetanus sequence and PF-03512676 adjuvant	Peptide	Phase 2b completed	Safe, immunogenic, and reduces the risk of recurrent HCMV infections in seropositive HCT recipients	[156,157]
**VBI-1501** **(VBI Vaccines)**	gB in eVLPs * and alum adjuvant	Recombinant and VLP	Phase 1 completed	Safe and immunogenic	[174,175]
**HB-101** **(Hookipa)**	replication-defective LCMV ** encoding HCMV gB and pp65	DNA, virus-vectored	Phase 2 ongoing	Phase 1: safe and immunogenic. Phase 2: (seronegative kidney transplant recipients) NA	[176,177]
**CMV-MVA Triplex** **(City of Hope)**	Modified MVA ***containing HCMVUL83 (pp65), UL123 (IE1), and UL122 (IE2)	DNA, virus-vectored	Phase 2 completed	Well-tolerated and immunogenic Reduces the risk of complications associated with HCMV infection by 50% in seropositive HCT recipients	[178]
**ASP0113 or VCL-CB01** **(Astellas)**	gB and pp65 with the adiuvants CRL1005 and benzalkonium chloride	DNA, plasmid-based	Phase2/Phase 3 completed	Well-tolerated; however, does not affect the rate of complications related to HCMV infection in HSCT recipients and renal transplant recipients	[179,180]
**mRNA-1647 (Moderna)**	six mRNAs, of which five encode the pentamer complex and one encodes gB	Self-replicating mRNA	Phase 2 and initiation of phase 3 announced in October 2021	NA and, according to the interim phase 2, well-tolerated	[181,182]

* eVLPs-enveloped virus-like particle, ** LCMV: lymphocytic choriomeningitis virus, and *** MVA: modified vaccinia Ankara.

**Table 5 ijms-23-03431-t005:** HSV vaccines.

Name of Vaccine (Developer)	Antigen/Other Vaccine Components	Type of Vaccine	Research Stage	Results	References
HSV529 (SanofiPasteur)	HSV-2 with deletions in *UL5* and *UL29*	Replication defective virus	Phase 1/2	Safe and immunogenic only in HSV-1 and 2 seronegative individuals	[197,200,201]
HSV-2 ΔgD-2 (X-Vax Technology)	HSV-2 ΔgD	Single -cycle virus	Preclinical, preparing for phase 1	Immunogenic in mice including HSV-1 seropositive and 100% protective in seronegative mice	[158,199]
RVx201 -ΔNLS (RationalVaccines)	HSV-2ΔNLS	Live attenuated	Preclinical, preparing for phase 1	Reduced recurrence rates in a small group of genital herpes patients studied without FDA approval	[202]
RVx1001 -HSV-1 VC2 (Rational Vaccines)	HSV-1 with deletion in gK (AA31-68)	Live attenuated	Preclinical, preparing for phase 1	Immunogenic in animals and protective against both HSV-2 genital infection (guinea pigs) and HSV-1 ocular infection (mice)	[198,199]
Gen-003 (Genocea)	gD2, infected cell polypeptide 4 (ICP4), and matrix-M2 adjuvant	Recombinant subunit	Phase 2 completed	Acceptable safety profile and immunogenic. Significant reduction of viral shedding and lesions rates in patients with genital herpes	[203,204]
HerpV (Agenus)	32 HSV peptides’ complex with HHSP 70 and QS21-adjuvant	Peptide	Phase 2 completed	Significant (17%) reduction in viral shedding in genital herpes patients	[205]
VCL-HB01 Vaxfectin (Vical)	gD, UL46/UL46, and Vaxfectin adjuvant	Plasmid-based DNA	Phase 2 completed (NCT02837575)	No difference from placebo in relapse rate in genital herpes patients tested	[194,206]
COR-1 (Admedus)	Codon-optimized gD2 and shortened gD2 fused with ubiquitin	DNA	Phase 2 completed	Herpes recurrence rate in the study group lower than baseline but the same as the placebo group	[195]
(UPenn, BioNTech)	HSV-2 mRNA coding gC2, gD2, gE2, and LNP *	mRNA	Preclinical, preparing for phase 1	Immunogenic and protective in mice and guinea pigs	[190]

* LNP- lipid nanoparticles.

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
