# Peer review of "40 Years after the Registration of Acyclovir: Do We Need New Anti-Herpetic Drugs?"

_ijms, 2022, doi:10.3390/ijms23073431_

Round 1

Reviewer 1 Report

Please include an index

Revise all references, the ones from clinical trials are not correct

Include figures to clarify concepts and info for sections 2 and 3.

Please include more references for the introduction such as  https://doi.org/10.3390/nano10071403

Al chemical structures should be in the same format ans chould be corrected (BCDV)

Figure 2 should have more quality

Author Response

Thank you for your review .

In reply to your comments:

  1. We included index of abbreviations
  2. We revise and corrected the references
  3. Figure 2 originally showing the mechanism of action of helicase-primase inhibitors has been expanded to include the mechanisms of action of other drugs such as nucleotide analogues nucleoside analogues and foscarnet. We have added Figure 3 showing the mechanisms of action of drugs against CMV
  4. We added the reference in the introduction
  5. We corrected the structural chemical formula of BCDV
  6. We improved quality of figures. The poor quality of the previous version was due to automatic compression when saving in a different format

Reviewer 2 Report

It is a laudable initiative to wonder whether, 40 years after the registration of acyclovir, do we need new antiherpetic drugs? Yet, the authors failed to provide an answer to this question. Such answer(s) seem mandatory to accept this paper for publication. The paper should profit from a thorough check-up of

1) style and grammar, i.e. page 1, line 19 and 20: rephrase, since cytomegalovirus is not just another important factor byt a human herpesvirus by itself

2) correct medicinal chemistry, i.e. page 11, line 408: CDV is not a pyrophosphate analogue but a nucleotide analogue, in fact an acyclic nucleoside phosphonate (ANP) analogue

3) BVDU (Brivudin) is ignored. Yet it is far more potent against VZV than acyclovir

Author Response

Thank you for your review . In reply to your comments:

  1. We have corrected the abstract (among other things, we've removed the part you mentioned) and we correct some wording in the text of the paper. We hope it will be sufficient. However, if the paper still contains a lot of language awkwardness we can ask the editors for an English edition
  2. We have carefully verified all chemical names used in the work
  3. We have added the missing description of BVDU in the nucleoside analogues section and in in Figure 1

Reviewer 3 Report

The MS sent me for estimation represents the biggest and most complex review on the etiotropic  agents directed against herpesvirus infections. The selection of data covers all effective anti-herpesvirus drugs, including the newest ones, illustrated by their chemical structure, mode of action including description of virus-specific targets, resistance development, results of clinical trials. The arrangement of the antivirals is excellent, as well.  All herpesviruses are included. The exhaustive and balanced presentation characterises very explicitly the enormous work of the authors of the MS. The same approach is applied when the vaccine preparatives are classified and discussed.

It would be more convenient to use the term biological response modifiers (BRM)  to represent not only immunomodulators, but also antioxidants and other agents acting only in vivo.  Moreover, the immunity problem is very sophisticated in herpesvirus infections. Corticosteroids by their effect against interferon bonding to cells could not be considered as antiviral agents – in opposite, they could be considered as virus-replication enhancers.  The mentioning of “unfavourable toxicity profile” of inhibitors of virus replication (p. 1, line 42), i.e. at the introduction of presentation of antivirals is mistaken, drug toxicity is presented in details during the exposition of the respective compounds. The series of papers on ellagitannins shows that these anti-herpetic compounds merit their inclusion in 3.3., based on their activity equal to ACV, effectivity versus ACV–resistant strains and their synergistic effect combined with ACV. The final chapter “5. Conclusion” is very short – it would be better if it is more complex and informative.

Author Response

Thank you very much for the good review of our paper. In reply to your comments:

  1. We used term biological response modifiers instead of immunomodulators.
  2. We have removed corticosteroids from the above group. We have discussed them separately

“A separate group are corticosteroids which, on the one hand, due to their immunosuppressive effect may worsen the course of viral diseases, on the other hand they are sometimes used to reduce inflammation.”

  1. We removed term “unfavorable toxicity profile” from page 1
  2. We have included a description of the effects of ellagitannins on HSV in section 3.3
  3. We have expanded the conclusions section